# *Mucuna pruriens* Administration Minimizes Neuroinflammation and Shows Anxiolytic, Antidepressant and Slimming Effects in Obese Rats

**DOI:** 10.3390/molecules25235559

**Published:** 2020-11-26

**Authors:** Renata leite Tavares, Maria Helena Araújo de Vasconcelos, Maria Letícia da Veiga Dutra, Aline Barbosa D’Oliveira, Marcos dos Santos Lima, Mirian Graciela da Silva Stiebbe Salvadori, Ramon de Alencar Pereira, Adriano Francisco Alves, Yuri Mangueira do Nascimento, Josean Fechine Tavares, Omar Guzman-Quevedo, Jailane de Souza Aquino

**Affiliations:** 1Experimental Nutrition Laboratory, Department of Nutrition, Federal University of Paraíba, Cidade Universitária, s/n-Castelo Branco III, João Pessoa 58051-085, Brazil; renataltav@gmail.com (R.l.T.); helenanutricionista@hotmail.com (M.H.A.d.V.); m.leticiavd@gmail.com (M.L.d.V.D.); allineolliveira99@gmail.com (A.B.D.); 2Department of Food Technology, Federal Institute of Sertão Pernambucano, Rod. BR 407 km 08, s/n-Jardim São Paulo, Petrolina 56314-522, Brazil; marcos.santos@ifsertao-pe.edu.br; 3Department of Psychology, Federal University of Paraíba, Cidade Universitária, s/n-Castelo Branco III, João Pessoa 58051-085, Brazil; mirian.salvadori@gmail.com; 4Pathology Laboratory, Department of Pathology, Federal University of Minas Gerais, Av. Pres. Antônio Carlos, n.6627-Pampulha, Belo Horizonte 31270-901, Brazil; ramon2alencar@gmail.com; 5Department of Physiology and Pathology, Federal University of Paraíba, Cidade Universitária, s/n-Castelo Branco III, João Pessoa 58051-085, Brazil; adrianofalves@gmail.com; 6Pharmaceutical Technology Laboratory, Department of Pharmaceutical Sciences, Federal University of Paraíba, Cidade Universitária, s/n-Castelo Branco III, João Pessoa 58051-085, Brazil; yurimangueira@ltf.ufpb.br (Y.M.d.N.); josean@ltf.ufpb.br (J.F.T.); 7Laboratory Neuronutrition and Metabolic Disorders, Higher Technological Institute of Tacámbaro, Av. Tecnológico 201, Tecario, Tacámbaro 61651, Mexico; gquevedomar@gmail.com

**Keywords:** cafeteria diet, bioactive compounds, interleukin 6, neurobehavior, obesity, weigh loss

## Abstract

This study evaluated the effect of *Mucuna pruriens* (MP) administration on neuroinflammation and behavioral and murinometric parameters in obese rats. Proximate composition, oligosaccharide and phenolic compound profile of MP were determined. Wistar adult male rats were randomized into healthy (HG) and obese group (OG). The HG consumed a control chow diet while OG consumed a cafeteria diet for eight weeks. Then, they were subdivided into: Healthy (HG); Healthy with MP administration (HGMP); Obese (OG); Obese with MP administration (OGMP), with the consumption of the respective diets remaining for another eight weeks, in addition to gavage with MP extract to supplemented groups (750 mg/kg weight). MP presented a composition rich in proteins and phenolic compounds, especially catechin, in addition to 1-kestose and levodopa. Supplementation reduced food intake, body weight, and thoracic and abdominal circumferences in obese rats. MP showed anxiolytic and antidepressant effects and reduced morphological damage and expression of interleukin 6 in the hippocampus of obese rats. MP treatment showed satietogenic, slimming, anxiolytic and antidepressant effects, besides to minimizing hippocampal neuroinflammation in obese rats. Our results demonstrated the potential anti-obesity of MP which are probably related to the high content of bioactive compounds present in this plant extract.

## 1. Introduction

Excess body weight affects thousands of people and has an increasing prevalence today [1]. The excessive accumulation of body fat, which characterizes obesity, is associated with chronic systemic inflammation present in this disease [2,3].

The hypertrophy and hyperplasia of adipose tissue which occurs in obese individuals leads to hypoxia and chronic inflammation, described as meta-inflammation, which is characterized by a low-grade and long-term inflammatory response, which affects several target systems and organs, affecting endocrine, metabolic, cardiovascular, pulmonary, gastrointestinal, hepatic, musculo-skeletal, hematological, reproductive, neuronal, and behavioral parameters, among others [4].

Although much has been studied about physical comorbidities associated with obesity, the relationship of this disease to mental health has only just begun to be explored in recent decades, relating it to psychiatric disorders such as anxiety, depression, binge eating and even changes in memory and cognition [5]. The cause for this relationship can be at the nervous system level, since obesity causes changes in neural circuits, neuroendocrine activity, metabolism and neurotransmitter and neurogenesis activity [6]. The hippocampus is the most studied region of the brain in depression related studies, and it also has relationship with anxiety disorders and neuroinflammation related to obesity [7,8,9]. IL-6 is one of the most important cytokines in chronic inflammation present in obesity, and is also one of the main cytokines secreted by different brain cells [10].

Prospecting for bioactive compounds which act to reduce neuroinflammation and can minimize the deleterious effects on the behavior of obese people can be used as a strategy in the treating obesity [2,6]. In this context, *Mucuna pruriens* (L.) DC (Fabaceae) is an herbal medicine traditionally used in Ayurveda which has a rich nutritional and phytochemical composition [11]. Previous studies have demonstrated antioxidant [12], anti-inflammatory [13] and anxiolytic [14] effects in healthy animal models. Such effects associated with the presence of Levodopa in *Mucuna pruriens* (MP) [14,15] can be promising in treating anxiety and depression associated with obesity, since it is precursor of dopamine which act directly in the control of hunger and satiety as well as in promoting well-being [5,16]. However, to our knowledge no study has been developed which has evaluated these effects of MP on obesity with a focus on weight loss and neurobehavioral parameters. In this sense, the present study aimed to evaluate the effect of MP administration on neuroinflammation and on behavioral and murinometric parameters in obese rats.

## 2. Results

### 2.1. Chemical Characterization of MP Extract

The MP extract showed a high percentage of protein and carbohydrates, a remarkable amount of ash and low fat and fiber content. High citric acid content was quantified, in addition to oligosaccharide 1-kestose and levodopa. The major phenolic compounds quantified were catechins, chlorogenic acid, trans-resveratrol and kaempferol 3-glucoside (Table 1). Appendix A present the results of the instrumental analyses performed.

### 2.2. Diet Consumption, Weight Monitoring and Murinometric Parameters

Throughout the obesity induction period (week 1 to 8), the animals that received a cafeteria diet (OG) had a significantly higher food intake (*p* ≤ 0.05) compared to rats who received the control chow diet (HG) (Figure 1A). A subsequent 14% reduction in food intake was observed when obese rats were treated with MP starting from the second week of administration in the OGMP group compared to OG (176.36 ± 11.90 g versus 205.11 ± 6.68 g, *p* ≤ 0.05), and maintained this difference until the end of the experiment. Moreover, no effect was observed when HG was treated with MP (Figure 1B).

Animals in the OG group already had higher body weight from the fourth week of the experiment than HG animals (286.29 ± 17.87 g versus 262.00 ± 22.09 g, *p* ≤ 0.05) (Figure 1C). MP administration caused a reduction (11%) in body weight after the thirteenth week of the experiment in OGMP rats compared to OG rats (393.75 ± 20.15 g versus 441.25 ± 21.74 g, *p* ≤ 0.05), maintaining this difference until the end of the experiment (*p* ≤ 0.05) (Figure 1D). Again, the MP administration had no effect on the HGMP.

It is important to highlight that the OGMP animals ended the study with similar values for food intake compared to HG (137 ± 2.44 g versus 133.33 ± 9.07 g, *p* > 0.05), as well as for weight gain (405.00 ± 29.58 g versus 352.00 ± 20.79 g, *p* > 0.05), indicating that treatment with MP was able to reverse hyperphagia and increased body weight caused by the cafeteria diet.

Corroborating the results of the body weight change, the murinometric parameters were also higher in the OG group compared to the OGMP for BMI (0.80 ± 0.04 g/cm^2^ versus 0.73 ± 0.03 g/cm^2^, *p* ≤ 0.05) and abdominal (19.08 ± 1.24 cm versus 16.50 ± 0.79 cm, *p* ≤ 0.05) and thoracic circumferences (16.92 ± 0.20 cm versus 15.93 ± 0.53 cm, *p* ≤ 0.05). However, there was no difference between these groups for the Lee Index (0.32 ± 0.01 versus 0.32 ± 0.01, *p* > 0.05) (Table 2).

### 2.3. Behavioral Parameters

In the elevated plus maze test the HG and HGMP animals remained longer in the center of the apparatus than the animals in the OG and OGMP groups (12.2 and 15.2 s versus 5.6 and 3.4 s, *p* ≤ 0.05) (Figure 2A). There was no difference in the time spent in the closed arms between the groups (HG, HGMP, OG and OGMP, respectively 279.2, 284.6, 288.4 and 296.1 s, *p* > 0.05) (Figure 2B), nor in the permanence of the rats in the open arms (0 s in all groups). However, treatment with MP normalized (OGMP) the reduction in the number of head dives observed in OG rats (2 dives versus 0 dives, *p* ≤ 0.05). Although there was no difference in the time spent in open arms, which is an important indicator of anxiety behavior, the increased number of head dives in OGMP animals may also indicate an anxiolytic effect of MP (Figure 2C).

OG animals showed less ambulation movements among all groups in the open field test (17.5 ambulation versus 29, 32 and 28.5 ambulation, *p* ≤ 0.05) (Figure 3A), indicating that MP administration reversed this decrease in OGMP. MP treated animals spent less time in grooming than obese animals (14.1 s versus 26.1 s, *p* ≤ 0.05) (Figure 3B). Healthy animals showed higher values than obese animals for rearing (number of rearing was 2.0 for HG; and 1.5 for HGMP versus 0.5 for OG; 0.5 and 1.0 for OGMP, *p* ≤ 0.05) (Figure 3C), regardless of treatment with MP. The highest number of fecal cakes was observed in the groups treated with MP (number of fecal cakes was 6.0 for HGMP and 5.0 for OGMP versus 3.0 for HG and 2.0 for OG, *p* ≤ 0.05) (Figure 3D). OG animals spent less time in center of the arena than HG, HGMP and OGMP (Figure 3E), and more time in the outer walls than HG, HGMP and OGMP (Figure 3F).

For the assessment of depressive behavior, the OG group showed less self-cleaning time in the splash test (31.3 s) versus 72.7 in HG, 65.7 in HGMP and 70.07 s in OGMP (*p* ≤ 0.05) (Figure 4). The OG group showed longer immobility (128.1 s) in the forced swimming test versus 57.3 in HG, 69.5 in HGMP and 94.3 s in OGMP (*p* ≤ 0.05) and shorter swimming (116.5 s) versus 198.3 in HG, 148.4 in HGMP and 180.9 s in OGMP (*p* ≤ 0.05) and diving times (1.3 s) versus 21.0 in HG, 25.7 in HGMP and 18.9 s in OGMP (*p* ≤ 0.05) (Figure 5). Animals which exhibit depressive-like behavior in the forced swimming test adopt an immobility posture, while animals which do not exhibit depressive-like behavior remain in motion (swimming or diving). Thus, the longer immobility time in OG indicated depressive behavior in these animals. It is important to highlight that the MP was able to reverse the depressive behavior, since the results were equal to those of the control group.

### 2.4. Histological and Immunohistochemical Analysis

Histological cuts of the brain in the region of the hippocampus stained in H&E showed neurons in the region of the toothed gyrus in normal histological aspects without alterations of the parenchyma or stroma (Figure 6A, B). Stromal alterations were observed in the OG, namely the presence of dilated vessels filled with blood (Figure 6C) in large numbers and extent indicating an acute inflammatory process, which was different from OGMP (Figure 6D), in which few dilated vessels were visualized when compared to O.

HG (Figure 6E) and HGMP (Figure 6F) animals had lower concentrations of IL-6 in the brain than OG (Figure 6G) and OGMP (Figure 6H). In turn, the OG group showed the highest IL-6 value per µ2 in brain, characterizing an inflammatory process at the level of the central nervous system in these animals (Figure 6I) and treatment with MP extract was able to reduce the expression of this pro inflammatory cytokine in the hippocampus of obese rats.

## 3. Discussion

A high content of phenolic compounds, such as catechin, chlorogenic acid, trans-resveratrol and kaempferol 3-glucoside, in addition to the oligosaccharide 1-kestose, was identified in the composition of the MP. Previous studies had identified the presence of saponins, tannins, anthraquinones, terpenoids and flavonoids in the alcoholic and ethanolic extract of MP [17] and quantified total phenolics (5.83 ± 0.57) and flavonoid content (13.25 ± 3.7) in the MP extract [18], but none of them determined the profile of phenolic compounds in the MP extract.

Few studies have investigated the fiber content in MP, with 9.1% being found by Handajani [19], from 5.9 to 12.1% by Vadivel and Janardhanan [20], and from 2.36 to 5.64% by Tavares et al. [11], but none of these studies evaluated the fractions of soluble and insoluble fibers, and even less oligosaccharides in this food matrix. Although the MP extract is not a source of fibers, the 1-kestose compound was quantified for the first time in MP and is an oligosaccharide with potential prebiotic action and may have influenced the reduction in food intake and consequently the weight loss of obese rats (OGMP), since oligosaccharides are prebiotics which contribute to a healthy gut microbiota leading to the production of short- chain fatty acids (SCFA), such as acetate, propionate and butyrate, which can increase the expression of hormones related to satiety, such as glucagon-like peptide-1 (GLP-1) and peptide YY (PYY) [21], or even regulate the appetite and decrease caloric intake via the hypothalamic nucleus [22].

Tochio et al. [23] demonstrated the potential prebiotic effect of oligosaccharide 1-kestose at a dose of 5 g/day by increasing the counts of *F. prausnitzii* and *Bifidobacterium* ssp. in human fecal samples. In our study, the animals consumed doses of 54.3 to 62.1 mg of 1-kestose/day through MP administration, which is equivalent to the dose of 5 g/day when converting the dose to humans [24]. Such effects may not have been as significant in healthy rats because the balanced microbiota of animals that consumed standard chow already guaranteed this psychobiotic balance, meaning the action of beneficial gut bacteria and their influence on the synthesis of brain neurotransmitters [25].

Treatment with MP extract, rich in nutritional and phytochemical composition, exhibited anxiolytic and antidepressant effects in the present study. In addition, it promoted weight loss and reduced food intake.

The reduction in food intake with consequent weight loss was reflected in the reduction of the BMI and chest and abdominal circumferences in the OGMP vs. OG, which proves the anti-obesity effect of MP. In our study, obesity was induced in rats through the cafeteria diet which has high palatability, low nutritional quality and high caloric density, and can induce hyperphagia in animals, resulting in greater fat deposits and weight gain [26]. It is important to highlight that the location of adipose tissue accumulation is of great importance for the pathophysiology of obesity, since the high amount of visceral adipose tissue demonstrated by the greater abdominal circumference tends to reflect a greater risk for developing metabolic disorders [27].

Although much is studied about physical and metabolic comorbidities associated with obesity, the relationship of this disease with mental disorders, such as anxiety and depression, has started to be studied more recently [5]. This relationship can be due to the systemic inflammation present in obesity, which causes changes in neural circuits, neuroendocrine activity and neurogenesis [6].

The elevated plus-maze test used to assess anxiety-related behavior is based on the natural exploratory behavior of rodents. Anxiety-like behavior is identified when the animal has a lower preference for open arms, and therefore a greater predilection for closed arms [28]. In the present study we observed that the cafeteria diet was anxiogenic, which was indicated by the shorter stay in the center in the apparatus and the lower number of head dives in animals in the OG. This result corroborates with the scientific literature, demonstrating that obesity alone is related to the development of anxiety [29,30]. MP can minimize the anxiogenesis characteristic of obesity, as observed in the present study by the increase in the number of head dives when animals of OGMP were in the center of the apparatus, demonstrating the natural exploratory behavior of rodents [31]. These results may indicate that the reduction of adipose tissue and consequent inflammation can contribute to the reduction of anxious-like behavior [32].

The OG animals showed less locomotor activity, less time spent in the center of the arena as well as more time spent in outer walls in the open field test, which is indicative of anxious behavior [29,33]. Likewise, these obese animals showed a greater response to stress, indicated by a longer grooming time. On the other hand, increased time spent in grooming behavior is related with anxiogenic states [32], as observed in the OG. For rearing, which indicates fearful behavior, healthy animals had fewer numbers than obese animals, regardless of treatment with MP (*p* ≤ 0.05). Despite the greater number of fecal cakes in the groups treated with MP, this parameter may not be indicative of anxiety, but of emotional reactivity [34], since the animals had been exposed to a new environment for the first time and this effect can be associated with the phytochemical composition of MP, which can increase peristaltic activity [35].

The anxiolytic effect of MP (250 to 750 mg/kg) was demonstrated in healthy rats which spent more time in the open arms in the elevated plus maze test and had a greater number of ambulation and rearing in the open field test. The anxiolytic activity of MP was associated with its possible action on the gabaergic and serotonergic systems, especially on agonists of the serotonergic receptors, which have an anxiety regulating effect [14]. It is important to note that the study cited [14] is the only one to date which related treatment with MP and anxiety, although it used healthy rats, without evaluating its effect on obesity.

MP is rich in levodopa and 5-hydroxytryptophane (5-HTP) and it can play an important role in the treatment of anxiety [14]. 5-HTP was not determined in our study, however, high levels of 5-HTP which is precursor of serotonin have been documented by several studies [15,36,37].

Levodopa is a compound present in several species belonging to the genus *Mucuna* [38]. Initially, the signals in the ^1^H-NMR spectrum referring to levodopa in the MP extract were assigned. The ABX-type hydrogenation characteristic of levodopa was confirmed in the ^1^H-NMR spectrum (Appendix A) with the following displacements δ_H_ 6.634 (d, *J* = 2.4 Hz, 1H), δ_H_ 6.628 (d, *J* = 8.0 Hz, 1H) and δ_H_ 6.493 (dd, *J* = 2.0 and 8.0 Hz, 1H), in addition, the possibility of these signals being attributed individually to other compounds present in the complex mixture of chemical constituents existing in a plant extract was ruled out by the COSY ^1^H-^1^H experiment (Appendix A) where it was observed a contour map referring to the ortho coupling existing in the aromatic ring of levodopa between the hydrogens δ_H_ 6.628 (d, *J* = 8.0 Hz, 1H) and δ_H_ 6.493 (dd, *J* = 2.0 and 8.0 Hz, 1H). The δ_H_ 6.493 signal was used to quantify the levodopa present in the MP extract, which does not overlap with any other signal in the ^1^H-NMR spectrum. Thus, the quantification of levodopa in the extract of MP revealed that this compound represents 14.08 ± 0.08% of the extract of MP (%*w*/*w*).

The presence of levodopa quantified in the composition of the MP (14.08%) in our study may also be involved in the antidepressant response demonstrated in OGMP via the action of the MP on the dopaminergic system, especially in neurons which have changes in the production of dopamine can induce depressive behavior [39]. In addition, previous studies have quantified in the protein fraction of MP appreciable amounts of the amino acids tyrosine and phenylalanine that are precursors of levodopa and may also have contributed to this mechanism of action [40,41].

The time spent on cleaning in the splash test indicates self-care and motivational behavior, so the shorter this time is, the greater the depressive behavior [42]. In the present study, splashing sucrose resulted in less self-cleaning time in animals in the OG, which indicates depressive behavior in these animals only, which in turn means that treatment with MP was able to reverse the depression that appears with obesity [43]. One hypothesis for these results is that the phenolic compounds and oligosaccharides quantified in MP in the present study may have contributed to this antidepressant effect through modulating the intestinal microbiota, which in turn had interaction with the central nervous system through the bloodstream, acting to reduce neuroinflammation and depressive behavior in animals [44].

Animals were observed for three behaviors in the forced swim test: swimming time, diving time and immobility time. Depressive-like behavior was identified in OG rats since the animals remained immobile longer and less time in movement (swimming and diving), indicating that the animals were not making an effort to escape the situation [45].

Only one study evaluated the antidepressant effect of MP [46], in which healthy animals treated with MP hydroalcoholic extract (100 and 200 mg/kg, orally) had shorter immobility time in the forced swim test, which proves its antidepressant effect. It is important to note that this previous study evaluated the antidepressant or anxiolytic effect of MP in eutrophic rats, meaning in non-obese rats. Treatment with MP in the present study increased the swimming time in OGMP rats, with an indication of the antidepressant effect of MP on obesity. The fact that if the animal is fat, it is more difficult that it may swim cannot be ignored, and once more, the slimming itself associated with MP phytochemical composition may have influenced this result.

On the other hand, the role of oligosaccharides in the microbiota-intestine-brain axis has been little investigated, as in the case of 1-kestose quantified in MP, despite the evidence that these prebiotics are capable of markedly modifying the relevant brain behavior and chemistry for anxious and depressive -like behaviors in rodents [47].

The relationship between phenolic compounds and behavior has been studied [31,39,48,49]. The majority phenolic compounds quantified in MP may have significantly contributed to the minimization of anxious and depressive-like behaviors in OGMP rats.

Catechins can modulate the noradrenergic system and activate gamma-aminobutyric acid receptors, presenting an anxiolytic effect via dopaminergic activation [31]. Chlorogenic acid had its anxiolytic and antidepressant effect demonstrated by Wang et al. [50], wherein they identified that 70% of this compound is metabolized by the intestinal microbiota, which strengthens the brain-intestine relationship, since diets which help in the balance of the intestinal microbiota can improve anxiety behavior through neurotransmission modulation [31].

In turn, resveratrol is one of the most potent polyphenols with antioxidant activity, can overcome the blood-brain barrier and cause antioxidant, anti-inflammatory and neuroprotective effects. Its antioxidant effect at the central level occurs through regulating oxidative stress of the mTOR pathway, modulating inflammatory cytokines, regulating serotonin and norepinephrine levels and activating brain-derived neurotrophic factor (BDNF) levels in the brain [48]. There are also reports that resveratrol’s anti-inflammatory action on behavioral changes may be the result of reduced pro-inflammatory cytokines, naturally increased in patients suffering from psychological stress [39]. It is important to note that the interplay between the mTOR pathway, oxidative stress and inflammation has a close relationship with hypothalamic regulation of body weight and food consumption [2,51]. Therefore, an interaction of resveratrol with these pathways could link metabolic effects to the anxiolytic benefits observed in this study. Kaempferol has neuroprotective activity by modulating the activity of gamma-aminobutyric acid receptors, being classified as anxiolytic [49]. The presence of these compounds in the composition of MP contributes to explain its anxiolytic and antidepressant effects, observed in the behavioral tests carried out in this study.

The induction of obesity caused an increase in dilated vessels, suggestive of inflammatory processes in the nervous system. Although many studies on obesity and neuroinflammation have only evaluated the hypothalamus, other areas of the brain have more recently been studied, such as the hippocampus [7]. As for behavior, the hippocampus is the most studied region of the brain in research related to depression, since it develops connections with emotion-related brain regions and regulates the hypothalamus-pituitary-adrenal axis, which makes it more susceptible to the development of stress and depression [8]. Still, anxiety disorders are also related to this area of the brain, as demonstrated by Mah, Szabuniewicz and Fiocco [9], who identified that chronic exposure to stress causes degeneration in the hippocampus. Administration with MP showed beneficial effects on the cerebral hippocampus of obese animals, since OGMP animals had less vasodilatation than OG.

The inflammation caused by the obesogenic diet at the brain level was able to increase IL-6 in the brain, as demonstrated by Dutheil et al. [52]. Low-grade inflammation in the central nervous system is also related to behavioral changes, such as anxiety and depression [53], as identified in the present study. In turn, MP administration was able to reduce brain inflammation mediated by IL-6, indicating its anti-inflammatory effect, which in turn may also be responsible for the identified behavioral outcomes. This means that MP administration presented a potential neuroprotective effect, especially due to its bioactive compounds that present anti-inflammatory actions, corroborating the relationship between these compounds and anxiolytic and anti-depressive behaviors. It should be noted that the administration dose used in this study (750 mg/kg of BW) can be adjusted for human application, using the human equivalent dose equation, obtaining the value of 120 mg/kg for humans [24], and constituting a dose which could be used in future translational studies.

## 4. Materials and Methods

### 4.1. Chemicals and External Standards for HPLC

External standards for 1-kestose, nystose, raffinose, glucose, fructose, rhamnose, maltose and for citric tartaric, lactic and acetic acids were purchased from Sigma-Aldrich (St. Louis, MO, USA). External standards for formic and succinic acids were purchased from Merck (Darmstadt, HE, Germany). External standards for procyanidin A2, epigallocatechin gallate, epicatechin gallate, kaempferol 3-glucoside, quercetin 3-rutinoside (rutin), quercetin 3-glucoside, myricetin, came from Extrasynthese (Genay, France). cis-Resveratrol and trans-resveratrol were obtained from Cayman Chemical Company (Ann Arbor, MI, USA). External standards of the phenolics: gallic acid, *p*-coumaric acid, chlorogenic acid, syringic acid, trans-caftaric acid, caffeic acid, hesperidin, naringenin, procyanidin B1, catechin, epicatechin and procyanidin B2, were purchased from Sigma-Aldrich. Acetonitrile, methanol, potassium persulfate, gallic acid and ethyl alcohol were obtained from Merck. The ultrapure water was obtained using a Marte Científica purification system (São Paulo, SP, Brazil). The chromogen 3,3′diaminobenzidine tetrachloride was purchased from Vector Laboratories (Burlingame, CA, USA). All antibodies used in this study were purchased from Abcam (Cambridge, MA, UK). All chemical reagents used in the experiments were of analytical grade.

### 4.2. Preparation of the MP Extract

The MP trees selected for seed collection were botanically identified and documented in the Lauro Pires Xavier Herbarium at the Federal University of Paraíba (UFPB). The MP seeds were obtained in the city of Alhandra, Paraíba, Brazil (latitude 07°26′19″ S × longitude 34°54′52″ W) and taken to the laboratory.

Afterwards, the seeds were rinsed in tap water, dried with absorbent paper at room temperature (± 26 °C), crushed by a mechanical grinder and sieved to obtain a fine powder (40 mesh), which was kept in a drying oven at 50 °C for 100 h. MP hydroalcoholic extract was obtained by adding distilled water and 95% ethanol (1:1, *v*/*v*), followed by homogenization, sieving and removal of solid residues and then subjected to a water bath at 45 °C for 24 h. The samples were frozen and then lyophilized at −48 °C with a pressure of 130 mmHg (Liotop, L101, São Carlos, Brazil) for 24 h to obtain the powdered MP extract [11,54].

### 4.3. Chemical Composition of MP Extract

#### 4.3.1. Proximate Composition and Total Energy Value

The moisture, ash, protein [55], lipid [56] and total, soluble and insoluble fiber [57] were determined. The total energy value was calculated according to the macronutrient levels (carbohydrates, proteins, and lipids). The experiment was carried out in triplicate.

#### 4.3.2. Quantification of Phenolic Profile, Organic Acids, Sugars and Oligosaccharides by High Performance Liquid Chromatography (HPLC)

Aqueous extracts were prepared to measure the contents of oligossacharides, organic acids, and sugars. Initially, 2 g of powdered MP extract were homogenized with 10 mL of ultrapure water for 10 min, centrifuged (3500× *g*, 10 min, 24 °C) and supernatant was filtered through a 0.45 μm nylon membrane (Millex Millipore, Barueri, SP, Brazil). Methanol extract was prepared to measure individual phenolics. First, 2 g of powdered MP extract were mixed with 10 mL of metanol:ultrapure water (70:30, *v*/*v*), treated with ultrasound (60 min, 35 kHz, 24 °C) and centrifuged (3500× *g*, 10 min, 24 °C). These procedures were repeated two times and collected supernatants were mixed and filtered (0.45 μm pore size).

The oligosaccharides nystose, 1-kestose and raffinose in HPLC/RID followed the method validated by Lima et al. [58]. The sample volume injected was 20 μL. The separation took place on a Synergi™ Hydro-RP C18 column with polar endcapping (150 × 4.6 mm, 4 μm) (Phenomenex, Torrance, CA, USA) at 35 °C. The solvent flow used was 0.7 mL. The gradient used was 0–8 min: 100% A; 8–9 min: 80% B; 9–12 min: 80% B; 12–13 min: 100% A; 13–20 min: 100% A (flowrate of 1.5 mL/min); in which solvent A was ultrapure water and solvent B was acetonitrile. The detection and quantification limits for all analyzed compounds were LOD < 0.042 g/L and LOQ < 0.109 g/L, respectively. All quantified oligosaccharides showed calibration curves with R2 > 0.998.

Sugars and organic acids were determined in HPLC-DAD-RID using the methodology validated by Coelho et al. [59]. An Agilent Hi-Plex H ion exchange column (300 × 7.7 mm, internal particles 8.0 μm) was used (Agilent Technologies, Santa Clara, CA, USA). The sample volume injected was 10 μL and the solvent flow was 0.6 mL/min. The mobile phase was a 4 mmol/L H_2_SO_4_ solution. The column oven temperature was maintained at 70 °C. Organic acids were detected in a DAD 210 nm, and sugars using a RID. All quantified sugars and acids showed calibration curves with R^2^ > 0.995. The limits of detection and quantification for all evaluated compounds were LOD < 0.042 g/L and LOQ < 0.131 g/L, respectively.

The individual phenolics was determined in RP-HPLC/DAD according to the method validated by Padilha et al. [60] with adaptations by Dutra et al. [61]. The column used was Reverse-Phase (RP) Zorbax Eclipse Plus C18 (100 × 4.6 mm, 3.5 μm) with a Zorbax C18 (12.6 × 4.6 mm, 5 μm) (Agilent Technology, Santa clara, CA, USA) pre-column. The injection volume was 20 µL of the sample and the oven temperature was maintained at 35 °C. The solvent flow was 0.8 mL/min. The gradient used in the separation was 0–5 min: 5% B; 5–14 min: 23% B; 14–30 min: 50% B; 30–33 min: 80% B, in which solvent A is a phosphoric acid solution (pH 2.0) and solvent B is methanol acidified with H_3_PO_4_ 0.5%. Compound detection occurred by comparison with the external standards. All quantified phenolics showed calibration curves with R^2^ > 0.997. The detection and quantification limits (LOD and LOQ) for all analyzed compounds were LOD < 0.17 mg/L and LOQ < 0.38 mg/L. All analyses by high performance liquid chromatography (HPLC) were performed using an Agilent 1260 Infinity LC chromatography system (Agilent Technologies) coupled to diode array detector (DAD) (model G1315D) and a refractive index detector (RID) (model G1362A). The experiment was carried out in triplicate.

#### 4.3.3. Quantification of Levodopa by Nuclear Magnetic Resonance (^1^H-NMR)

The Acquisition of the ^1^H-NMR spectrum was performed on a Bruker Ascent 400 instrument operating at 400 MHz for ^1^H-NMR and at 100 MHz for ^13^C-NMR (Bruker, Billerica, MA, USA). The following parameters were used to obtain the spectra of solvent ^1^H-NMR: DMSO-*d*_6_; temperature: 26 °C; number of scans: 16; receiver gain: auto; acquisition time: 4s. For quantification, TopSpin Eretic 2 was used, which was calibrated using gallic acid (Sigma, Darmstadt, Germany) (118.53 mMol/L) used as an external standard, using the same parameters described above. In the quantification, the peak area was used, which had the initial and final points of integration done manually. The concentration of MP extract used to perform the tests was 23.67 mg/mL. The experiment was carried out in triplicate. The value of levodopa present in the extract of MP is expressed in %weight of levodopa ± standard deviation/weight of the MP seed extract (%*w*/*w*).

### 4.4. Study Design

The experimental protocol started after approval of the project by the Ethics Committee on the Use of Laboratory Animals at the Federal University of Paraíba (CEUA-UFPB), under protocol number 4657230418, and followed the guidelines of the National Council for The Control of Animal Experimentation (Conselho Nacional de Controle de Experimentação Animal–CONCEA, Brazil) and Institute of Laboratory Animal Resources [62]. The study design is described in Figure 7.

Thirty-two (32) male Wistar rats aged ± 40 days old were used in this study, kept under standard lighting conditions (12/12-h light/dark cycle, light off at 7 pm), humidity (55 ± 10%) and temperature (22 ± 2 °C). The animals were kept in collective polypropylene cages measuring 41 × 34 × 16 cm (two animals per cage), where they received filtered water and diets ad libitum. The rats were initially randomized into two groups for the first eight weeks (obesity induction) of the experiment: healthy group (HG, *n* = 16) and obese group (OG, *n* = 16). Next, the rats were randomly assigned into four groups for the following eight weeks: healthy group (HG, *n* = 8); healthy group with MP administration (HGMP, *n* = 8); obese group (OG, *n* = 8); obese group with MP administration (OGMP, *n* = 8).

The cafeteria diet was chosen because it is the best diet that mimics the dietary lifestyle in society these days. A cafeteria diet was offered for sixteen weeks for the two groups of obese animals (OG and OGMP), previously validated for obesity induction, being offered a control chow diet (Presence, Paulínia, Brazil) associated with processed foods such as sausages, french fries, treats, cookies, stuffed cakes, and corn snacks, among others, considered hypercaloric and, sources of salt, fat and sugars [63,64]. The animals in the HG and HGMP groups only received the control chow diet (Presence, Paulínia, Brazil). Throughout the entire study, the standard diet offered to rats has 308 kcal/100 g, 45% carbohydrates, 23% protein, 4% lipids, while the cafeteria diet has 407 kcal/100 g, 48% carbohydrates, 14% proteins and 38% lipids, rich in refined sugars, salt, food additives, trans and saturated fat, as well as being low in fiber.

After the obesity induction period, the animals belonging to the groups with MP administration (HGMP and OGMP) received 750 mg/kg of MP extract diluted, in 1 mL of distilled water daily for eight weeks via gavage [14]. Animals from groups not treated with MP (HG and OG) received gavage with 1 mL of distilled water. The MP extract dose (750 mg/kg) was established based on a pilot study and on the study conducted by Rai et al. [14] who evaluated the anxiolytic effect of MP in healthy rats. The chosen dose was also considered safe from previous toxicity studies [65,66].

Daily food intake was assessed daily at the same time (10 pm), being represented by the difference in grams between the offered and the residual diet [64]. The animals’ body weight was checked weekly at the same time (9 p.m.) by directly weighing each animal on an electronic scale (Prix 3/1, Toledo, São Bernardo do Campo, Brazil). All rats survived until the end of this study, with no adverse events.

### 4.5. Evaluation of Behavioral Parameters

The animals’ behaviors related to anxiety and depression were evaluated at the end of the 16 weeks of study, each test being performed on a specific day. The elevated plus maze test and the open field test were used to assess the rats’ anxiety behavior. In addition, the forced swimming test and the splash test were performed to evaluate the behavior related to depression in the rats.

#### 4.5.1. Anxiety-Like Behavior in Rats

Tests in the elevated plus maze and in the open field were performed in the last week of treatment. The elevated plus maze test used has two open and opposite arms, measuring 50 × 10 cm, and two arms closed on its three external faces by walls 40 cm high, with platforms with the same size as the open arms, crossing them perpendicularly, which delimits a central area of 100 cm^2^. The apparatus is elevated 50 cm from the ground and has lighting from a 60-watt white lamp, located 120 cm above the apparatus. The animal was positioned on the central platform facing one of the open arms and observed for five minutes for the following parameters: number of entries in the open arms; number of entries in closed arms; length of stay in open arms; length of stay in closed arms; length of stay in the central area; and head diving. Thus, when a rat is exposed to the elevated plus maze apparatus for the first time, it shows signs of conflict, avoidance and escape, and an animal which exhibits an anxiety behavior enters less often and remains for less time in the open arms [67,68].

The open field was used on the second test, which consists of a circular acrylic arena of 1 m diameter, 50 cm high side walls and a white floor with black lines consisting of two concentric circles divided by radial lines forming areas of 20 cm^2^. The animal was positioned in the center of the apparatus and observed for five minutes for the following behaviors: ambulation (recorded by the number of segments crossed with four legs by the animal), grooming time (self-cleaning), rearing number (behavior in which the animal only balances on its hind legs, without touching the walls), defecation (number of fecal cakes present in the appliance), time spent in center of the arena and time spent in close to the outer walls [69,70].

The devices were thoroughly cleaned with 10% alcohol at each animal change in both tests to avoid interference in the tests. The records of each test were performed using an IP camera (D-Link brand, model IP2P Wireless, Shenzhen, China) attached to a support installed in each of the devices.

#### 4.5.2. Depression-Like Behavior in Rats

Tests which assess the depressive behavior in rats were performed in the last week of treatment. The splash test consists of squirting 200 µL of 10% sucrose solution into the animal’s snout. Due to its viscosity, the sucrose solution impregnates the skin and hair of the rat and the animals initiate self-cleaning behavior which has the time measured for five minutes after applying the solution. Depressive-like behavior in the animal is indicated if the self-cleaning time is shorter compared to the control [71].

In the forced swimming test, the rats were placed in a cylindrical polyvinyl chloride plastic tank 25 cm diameter and 70 cm deep, filled to 60% with water at 25 ± 2 °C. This amount of water prevents the animal from resting on their feet at the bottom of the cylinder and escaping from the top edge. The animals had their behavior monitored for five minutes. At the beginning of the test, the animals swim and make movements to climb the inner walls of the cylinder, and over time they adopt a posture of immobility, recognized as a depressive-like behavior. The times that the animals remain swimming, immobile and immersed were quantified. The water was changed, and the apparatus was cleaned with 10% alcohol between the test with one animal and another [28,72]. The records of each test were performed using an IP camera (D-Link brand, model IP2P Wireless, Shenzhen, China).

### 4.6. Murinometric Parameters, Euthanasia and Tissue Samples

The animals were fasted for 8 h at the end of the 16 weeks of the experiment and had their murinometric parameters measured, such as: chest circumference, abdominal circumference, body weight and length using an inextensible measuring tape, all measured in cm. The body weight (g) was divided by the length squared (cm^2^) to calculate the body mass index (BMI) [73], and the Lee Index was calculated by the cubed root of the body weight (g) divided by the length (cm), according by ^3^√Body weight/body length [74]. Rats with BMI and Lee index with statistically significant differences in relation to control are considered obese [75].

The animals were euthanized by beheading, according to the ethical principles of National Council for The Control of Animal Experimentation (Conselho Nacional de Controle de Experimentação Animal–CONCEA, Brazil) and Institute of [62]. Brain tissues were collected for histological analysis.

### 4.7. Histological and Immunohistochemical Analysis

Brain samples of each animal were washed with saline, fixed in 10% buffered formalin and stored in coded containers. Ten semi-serial cuts of 5 µm thick were obtained from the paraffin embedded material, following a cross-sectional plane to the analyzed organ of each animal. The obtained slides were stained using the hematoxylin and eosin (H & E) technique, and the assembly was performed between lamina and laminula with synthetic resin (Entellan^®^-Merck, Darmstadt, HE, Germany) for analysis in increasing lenses and photographed at 100× total magnification under an optical microscope (Motic BA 200, Kowloon, Hong Kong). The structural architectures of the brain and the presence, characteristic and intensity of possible inflammatory infiltrates were evaluated in these analyzes [28].

For immunohistochemical staining, a peroxidase blocker with 3% hydrogen peroxide for 5 min was used, incubated with anti-IL-6 antibody diluted 1:100 (abc324; Abcam Cambridge, UK) for IL-6 for 60 min. room temperature. Subsequently, the samples were incubated with secondary antibodies conjugated with horseradish peroxidase polymer for 10 min at room temperature. The samples (hippocampus regions) were then treated with a chromogen, 3,3′diaminobenzidine tetrachloride (Vector Laboratories) for 3 min at room temperature and contrasted with hematoxylin for 5 min. The images were captured using an Eclipse 80i microscope (Nikon, Tokyo, Japan) at 400× total magnification. Finally, for morphometric analysis of the positivity area for immunohistochemical reaction, 20 random images were captured using a microcamera attached to the microscope, in each type of mark and organ studied. The immunohistochemical marking area was calculated using algorithms built in the KS300 software (Zeiss, Zaventem, BE, Belgium). In each image, all pixels with shades of brown (positive immunohistochemical marking) were selected for the creation of a binary image, digital processing, and area calculation in μm^2^ [76,77].

### 4.8. Statistical Analysis

The data of body weight, murinometric parameters and food intake data were presented as mean and standard deviation of the mean and tested for normality by the Kolmogorov-Smirnov test. Food intake and body weight were then submitted to analysis of variance of repeated measures (two-way ANOVA), while murinometric parameters as submitted to one-way ANOVA. The Tukey post hoc test was performed with a significance level of 5% (*p* ≤ 0.05) when there was a difference between the variables. In two-way ANOVA tests, factor A was the condition of the animals (normal and obese rat), and factor B was the treatment administered (distilled water or MP).

The results of the behavioral tests were presented as median (minimum and maximum) and evaluated by the Kruskal-Wallis test. The Dunn post hoc test was performed with a significance level of 5% (*p* ≤ 0.05) when there was a difference between the variables. The analysis was performed using the Instat 3.0 software, with no blinding (GraphPad Inc., San Diego, CA, USA).

## 5. Conclusions

This was the first study associating the effect of MP with the relationship between behavior and obesity. The beneficial effects on weight loss, appetite, neuroinflammation and behavioral parameters found in this study may have been promoted by the rich content of bioactive compounds present in MP, specifically by oligosaccharides, levodopa and phenolic compounds. This study revealed that the administration of MP is a promising strategy in the fight against obesity, although more studies should be developed in order to test the metabolic effects on the pathophysiology of obesity, either from this or other doses at different times of administration, with focus on elucidating the mechanisms of action of bioactive compounds of MP on the gut-brain axis.

## Figures and Tables

**Figure 1 molecules-25-05559-f001:**
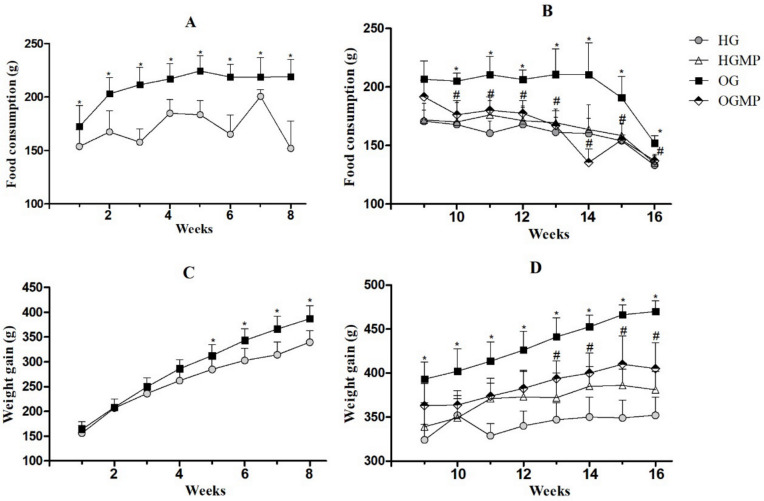
Food consumption (**A**,**B**) and weight gain (**C**,**D**) in healthy and obese rats in the pre- and post-administration period with MP seed extract. Data presented as mean and standard deviation of the mean. Two-way ANOVA, *p* ≤ 0.05 Tukey’s post hoc test. * differed from HG; # differed from OG.

**Figure 2 molecules-25-05559-f002:**
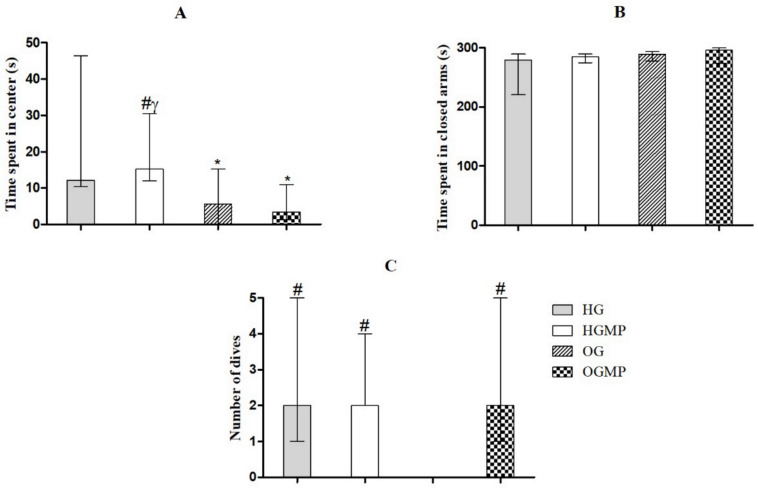
Results of the elevated plus-maze test regarding the time spent in the center of the apparatus (**A**), time spent in the closed arm (**B**) and number of dives (**C**) of healthy and obese rats treated or not with MP seed extract. Data presented as median (minimum and maximum). Kruskal-Wallis test, Dunn post-hoc (*p* ≤ 0.05). * differed from HG; # differed from OG.; γ differed from OGMP.

**Figure 3 molecules-25-05559-f003:**
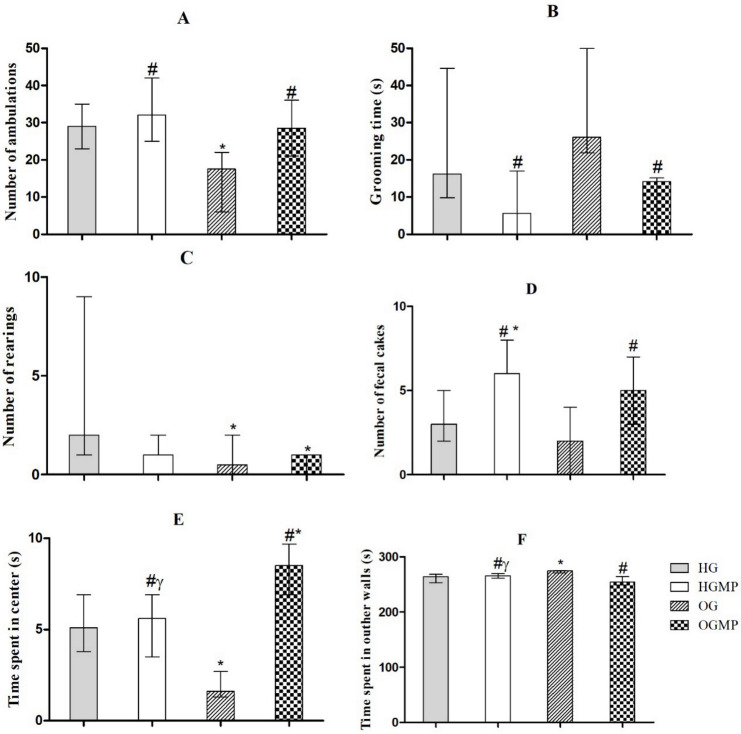
Results of the open field test regarding the number of ambulation (**A**), grooming time (**B**), rearing number (**C**), number of fecal cakes (**D**), time of spent in center (**E**) and time spent in outher walls (**F**) evaluated in healthy and obese rats treated or not with MP seed extract. Data presented as median (minimum and maximum). Kruskal-Wallis test, Dunn post-hoc (*p* ≤ 0.05). * differed from HG; # differed from OG; γ differed from OGMP.

**Figure 4 molecules-25-05559-f004:**
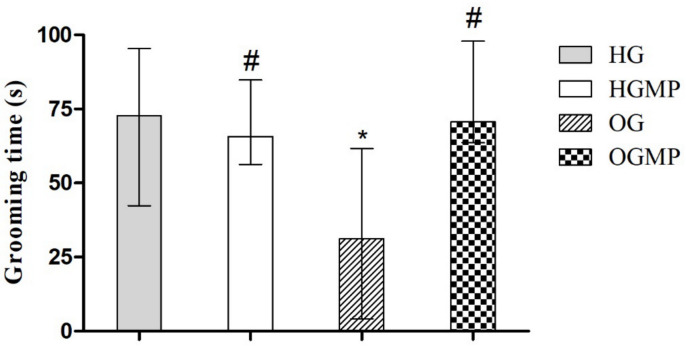
Results of the splash test (grooming time in seconds) in healthy and obese rats treated or not with MP seed extract. Data presented as median (minimum and maximum). Kruskal-Wallis test, Dunn post-hoc (*p ≤* 0.05). * differed from HG; # differed from OG.

**Figure 5 molecules-25-05559-f005:**
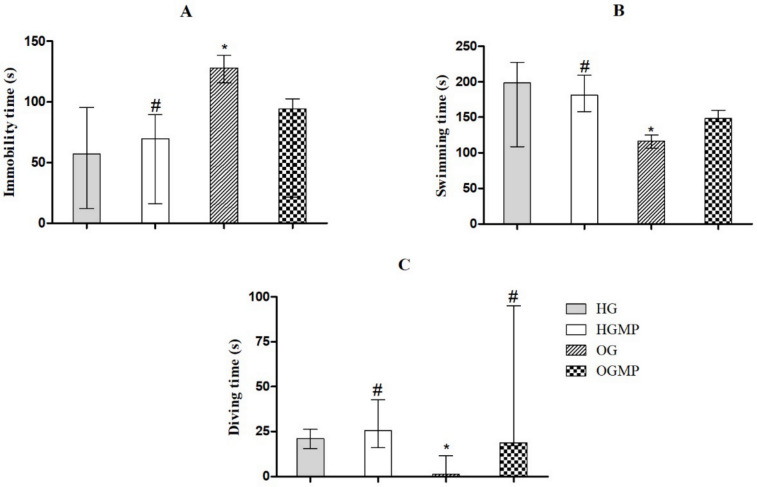
Results of the forced swim test regarding the immobility time (**A**), swimming time (**B**) and diving time (**C**) evaluated of healthy and obese rats treated or not with MP seed extract. Data presented as median (minimum and maximum). Kruskal-Wallis test and Dunn post-hoc (*p* ≤ 0.05). * differed from HG; # differed from OG.

**Figure 6 molecules-25-05559-f006:**
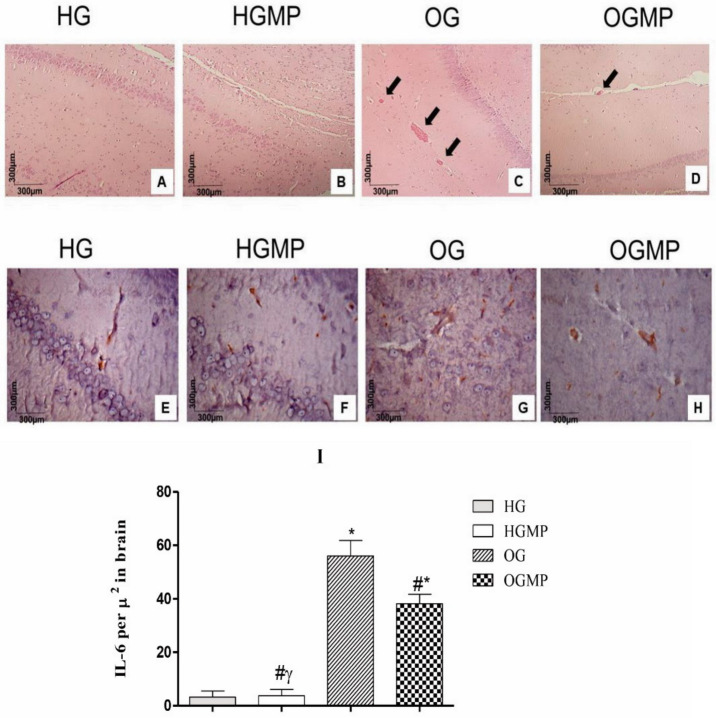
Histology (**A**–**D**), immunohistochemical (**E**–**I**) analysis of the hippocampus of healthy and obese rats treated or not with MP. The arrow indicates dilated vessels. Data presented as mean and standard deviation of the mean. Two-way ANOVA, *p* ≤ 0.05 Tukey’s post hoc test. * differed from HG; # differed from OG; γ differed from OGMP.

**Figure 7 molecules-25-05559-f007:**
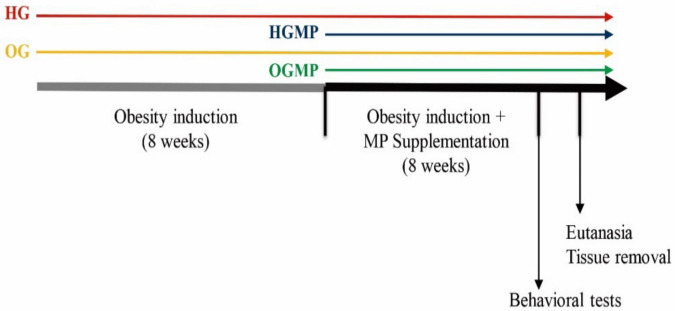
The study design timeline and steps of the biological assay. HG = healthy group; OG = obese group; HGMP = Healthy group with MP administration; OGMP = Obese group with MP administration.

**Table 1 molecules-25-05559-t001:** Chemical composition of hydroalcoholic extract of Mucuna pruriens (MP) seeds.

Parameters	Mean and Standard Deviation
Moisture (g/100 g)	3.34 ± 0.60
Ash (g/100 g)	2.63 ± 0.26
Protein (g/100 g)	7.11 ± 0.30
Lipid (g/100 g)	0.41 ± 0.05
Carbohydrate (g/100 g)	86.63 ± 0.27
Monosaccharide and disaccharide (mg/g)	
Fructose	12.65 ± 0.03
Glucose	17.62 ± 0.05
Maltose	7.74 ± 0.02
Dietary fiber (g/100 g)	
IDF	1.01 ± 0.54
SDF	1.42 ± 0.02
TDF	2.43
IDF:SDF ratio	0.71
Total energy value (kJ/100 g)	1579.86 ± 7.22
Total energy value (kcal/100 g)	377.59 ± 1.73
Oligosaccharide (mg/100 g)	
1-Kestose	20.70
Nystose	n.d.
Raffinose	n.d.
Organic acids (mg/100 g)	
Citric	31.76 ± 0.06
Tartaric	8.57 ± 0.04
Formic	18.15 ± 0.08
Levodopa (%)	14.08 ± 0.08
Phenolic compounds (mg/100 g)	
Flavanol	
Catechin	57.83 ± 0.06
Procyanidin B1	16.44 ± 0.09
Procyanidin B2	18.41 ± 0.08
Flavonol	
Quercitin 3-Glucoside	13.00 ± 0.05
Kaempferol 3-Glucoside	19.34 ± 0.04
Total flavonoids	125.02
Phenolic acids	
Chlorogenic acid	49.32 ± 0.06
Stilbenes	
Trans-resveratrol	21.45 ± 0.05
Total non-flavonoids	70.77
Total phenolic compounds	195.79

Data presented as mean and standard deviation of the mean. n.d. = not detected.; SDF = soluble dietary fibers; IDF = insoluble dietary fibers TDF = total dietary fibers.

**Table 2 molecules-25-05559-t002:** Murinometric parameters of healthy and obese rats after MP seed extract treatment.

Parameters	Groups			
	HG	HGMP	OG	OGMP
Body length (cm)	23.2 ± 0.91	22.63 ± 0.64 # γ	24.75 ± 0.88 *	24.0 ± 0.55
Abdominal circumference (cm)	15.33 ± 0.82	15.78 ± 0.67 #	19.08 ± 1.24 *	16.5 ± 0.79 #
Thoracic circumference (cm)	14.67 ± 0.68	14.75 ± 0.6 # γ	16.92 ± 0.2 *	15.93 ± 0.53 # *
BMI (g/cm^2^)	0.71 ± 0.04	0.71 ± 0.03 #	0.80 ± 0.04 *	0.73 ± 0.03 #
Lee Index	0.31 ± 0.01	0.32 ± 0.01	0.32 ± 0.01	0.32 ± 0.01

Data presented as mean and standard deviation of the mean. One-way ANOVA, *p* ≤ 0.05 Tukey’s post hoc test. # differed from OG; * differed from HG; γ differed from OGMP.

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
