# Peer review of "Mucuna pruriens Administration Minimizes Neuroinflammation and Shows Anxiolytic, Antidepressant and Slimming Effects in Obese Rats"

_molecules, 2020, doi:10.3390/molecules25235559_

Round 1

Reviewer 1 Report

The paper describes effects of Mucuna pruriens administration on neuroinflammation and an anxiolytic, antidepressant, and slimming behaviour in obese rats. The main claims of the paper are properly placed in the context of the previous literature. The experimental data support the claims. An abstract represents the achievements of work. Reviewer recommends the paper to further publishing process after minor revision.

Main points:

  1. The conclusion from page 9

“The anxiolytic effect of MP was evaluated in healthy rats by Rai et al.[14] , and it was observed that the animals that received gavage with MP flour in doses of 250 to 750 mg/kg spent more time in the open arms in the elevated plus maze test and had a greater number of ambulation and rearing in the open field test, demonstrating the anxiolytic activity of MP through its possible action on the gabaergic and serotonergic systems and on 5-hydroxytryptophan (5-HTP) agonists, which have an anxiety regulating effect.”

I wonder, what are 5-HTP agonists? 5-HTP is an amino acid, precursor of serotonin 5-HT. Based on [Pranzatelli MR. Effect of chronic treatment with 5-hydroxytryptophan on cortical serotonin receptors in the rat. Clin Neuropharmacol. 1988 Jun;11(3):257-62. doi: 10.1097/00002826-198806000-00008. PMID: 3261199]:

“In rats treated with high doses of 5-HTP (50-200 mg/kg), cortical 5-HT2 (-20%) and 5-HT1 (-11%) sites were downregulated without altered receptor affinity, but only the changes in 5-HT2 sites were significant. The differential effects of high-dose 5-HTP on 5-HT receptors suggest that 5-HT2 receptor downregulation may be relevant either to the antimyoclonic effect of chronic 5-HTP therapy in posthypoxic myoclonus or to development of tolerance”.

In addition to this misunderstanding, sentence is too long.

Minor points

Abstract:

Line 37-39.

MP treatment showed a sacietogenic and slimming effects and reduced anxious and depressive like behavior, as well as hippocampal neuroinflammation in obese rats, probably related to the high content of bioactive compounds in MP, demonstrating its potential use to combat obesity.

Comment: sentence is too long and embroiled

Introduction

Line 61-63

Looking for mechanisms involving reduction of inflammation at the central nervous system level, can minimize the deleterious effects on the behavior of obese individuals and might be a strategy in treating obesity.

Comment: something is missing

Results:

Table 1:

There is no information about 5-HTP level in MP. Please complete the table

Author Response

João Pessoa/Paraíba - Brazil, november/2020.               

REVIEWER 1

Dear Reviewer,

We are pleased to send you the revised version (R1) of the paper “Mucuna pruriens administration minimizes neuroinflammation and shows anxiolytic, antidepressant and slimming effects in obese rats” after carefully considering the comments of the Reviewers and the Editor. All suggestions and requirements were considered to improve the general scientific quality of the manuscript. All the manuscript was revised in order to fit its format to the standard required by Molecules. The changes in the text are highlighted using the “track changes” function in Microsoft word regarding each Reviewer’s comments. As requested, a clear list of the responses to the Reviewers’ comments (including line numbers that indicate where the changes have been made) is presented.

Comments

The paper describes effects of Mucuna pruriens administration on neuroinflammation and an anxiolytic, antidepressant, and slimming behaviour in obese rats. The main claims of the paper are properly placed in the context of the previous literature. The experimental data support the claims. An abstract represents the achievements of work. Reviewer recommends the paper to further publishing process after minor revision.

Answer

We appreciate all the reviewer' suggestions and the recognition of the merit of this manuscript.

Main points:

The conclusion from page 9

“The anxiolytic effect of MP was evaluated in healthy rats by Rai et al.[14] , and it was observed that the animals that received gavage with MP in doses of 250 to 750 mg/kg spent more time in the open arms in the elevated plus maze test and had a greater number of ambulation and rearing in the open field test, demonstrating the anxiolytic activity of MP through its possible action on the gabaergic and serotonergic systems and on 5-hydroxytryptophan (5-HTP) agonists, which have an anxiety regulating effect.”

I wonder, what are 5-HTP agonists? 5-HTP is an amino acid, precursor of serotonin 5-HT. Based on [Pranzatelli MR. Effect of chronic treatment with 5-hydroxytryptophan on cortical serotonin receptors in the rat. Clin Neuropharmacol. 1988 Jun;11(3):257-62. doi: 10.1097/00002826-198806000-00008. PMID: 3261199]:

“In rats treated with high doses of 5-HTP (50-200 mg/kg), cortical 5-HT2 (-20%) and 5-HT1 (-11%) sites were downregulated without altered receptor affinity, but only the changes in 5-HT2 sites were significant. The differential effects of high-dose 5-HTP on 5-HT receptors suggest that 5-HT2 receptor downregulation may be relevant either to the antimyoclonic effect of chronic 5-HTP therapy in posthypoxic myoclonus or to development of tolerance”.

In addition to this misunderstanding, sentence is too long.

Answer

We appreciate the correction and we rewrote the sentence according to the aspects raised by the reviewer (please see lines 263-267):

“The anxiolytic effect of MP (250 to 750 mg /kg) was demonstrated in healthy rats which spent more time in the open arms in the elevated plus maze test and had a greater number of ambulation and rearing in the open field test. The anxiolytic activity of MP was associated with its possible action on the gabaergic and serotonergic systems, especially on agonists of the serotonergic receptors, which have an anxiety regulating effect [14].”

Minor points

Comments

Abstract:

Line 37-39. Comment: sentence is too long and embroiled.

MP treatment showed a sacietogenic and slimming effects and reduced anxious and depressive like behavior, as well as hippocampal neuroinflammation in obese rats, probably related to the high content of bioactive compounds in MP, demonstrating its potential use to combat obesity.

Answer

We appreciate the correction and we rewrote the sentence according to the aspects raised by the reviewer (please see Abstract - lines 40-43):

“MP treatment showed satietogenic, slimming, anxiolytic and antidepressant effects, besides to minimizing hippocampal neuroinflammation in obese rats. Our results demonstrated the potential anti-obesity of MP which are probably related to the high content of bioactive compounds present in this plant extract.”

Comments

Introduction

Line 61-63 Comment: something is missing

Looking for mechanisms involving reduction of inflammation at the central nervous system level, can minimize the deleterious effects on the behavior of obese individuals and might be a strategy in treating obesity.

Answer

We appreciate the correction and we rewrote the sentence according to the aspects raised by the reviewer (please see page 2 - lines 65-67):

“Prospecting for bioactive compounds which act to reduce neuroinflammation and can minimize the deleterious effects on the behavior of obese people can be used as a strategy in the treating obesity”

Comments

Results:

Table 1:

There is no information about 5-HTP level in MP. Please complete the table

Answer

We did not quantify 5- HTP in our study and, therefore, this result is not shown in Table 1. We also made this information clearer in the discussion (please see page 10- lines 270-272).

Best regards,

The authors

Reviewer 2 Report

Manuscript Number:       molecules-985934

Title:                             Mucuna pruriens administration minimizes neuroinflammation and shows anxiolytic, antidepressant and slimming effects in obese rats

Comments to Authors

Recommendation:          Major revision

The article "Mucuna pruriens administration minimizes neuroinflammation and shows anxiolytic, antidepressant and slimming effects in obese rats” has the goal to evaluate the effect of Mucuna pruriens administration on neuroinflammation and behavioural and murinometric parameters in obese rats. The proximate analysis of the chemical composition was performed, and oligosaccharide and phenolic compound profiles of M. pruriens were determined.

Although the interesting theme, some aspects of the introduction, experiments and discussion need considerable improvement.

In Introduction section, authors should provide the scientific explanation why the mentioned plant Mucuna pruriens was chosen to establish the targeted activities, as the authors were aware of other activities the mentioned plant was used in Ayurveda. Authors should provide the complete information citing the appropriate literature regarding the traditional use of the investigated plant, not stating only : “…a herbal medicine traditionally used in Ayurveda, but not for slimming purposes...”

Please, define what part of the plant was used for the tested extracts (legume is not precise terminology, meaning that the author should point if the semen or something else was used).

Please, give the scientifically explanation for the chosen dose applied in the experiment (750mg/kg)? Why only one dose was used?

The main concerns regarding the presented results refer to the authors’ analysis of the chemical composition of the used extract in the presented and performed experiments. Although the authors based their explanation of the exhibited effects on chemical profile of the extract used in investigation, in Material and Methods they did not provide the procedure of obtaining the extract, the methods used for the qualitative and quantitative chemical analysis of the investigated extracts. It is, at least, insufficient to say that the “lot was the same”, to give the literature from where the methodology was taken over. The subsection 4.1 could not be named….Nutritional characterization of MP, as the data presented in Results showed the presence of non-nutritional secondary metabolites, such as catechin, polyphenolic compounds etc. The whole preparation of the extracts has not been precisely, scientifically and accurately presented. Please, give the HPLC chromatogram, give the brief procedure for extract preparation, and for all methods used to determine the mentioned groups of the compounds present in the extracts. The presented tables in Results section did not give the information whether the presented results refer to plant material or extract (table1) – please, clarify

Please, explain how the extract was diluted to be administrated in all experiments, taking into account that the extract contained quite different compounds regarding their solubility.

Please, give the data whether all animals survived all 8 weeks of the experiment.

Some minor corrections that should be done:

English should be corrected: proximal / proximate; sacietogenic / satietogenic: page 2 lines 61, 62; page 3, lines 83,84, page10, line289, should be corrected, and etc…. The Results and Discussion sections should be written in Past Tense. Please, check throughout the text, and prior sending the corrected version, consult the English language native speaker

All figures are of poor quality, the quality should be improved.

The numbers of subsections should be corrected – there is repeating – 2.3….

Page 6, lines 148-156, please, specify the link between the given values and the corresponding investigated groups

When the plant has been mentioned for the first time, the full Latin name should be provided: Latin name, author abbreviation, and Family, while full name should be in Italic. Afterwards, authors might switch to MP or M. pririens. It refers to Bacterial strains as well (page 8, line 206).

The authors should provide the name of the expert who identified the plant material.

Author Response

João Pessoa/Paraíba - Brazil, november/2020.               

REVIEWER 2

Dear Reviewer,

We are pleased to send you the revised version (R1) of the paper “Mucuna pruriens administration minimizes neuroinflammation and shows anxiolytic, antidepressant and slimming effects in obese rats” after carefully considering the comments of the Reviewers and the Editor. All suggestions and requirements were considered to improve the general scientific quality of the manuscript. All the manuscript was revised in order to fit its format to the standard required by Molecules. The changes in the text are highlighted using the “track changes” function in Microsoft word regarding each Reviewer’s comments. As requested, a clear list of the responses to the Reviewers’ comments (including line numbers that indicate where the changes have been made) is presented.

Comments

The article "Mucuna pruriens administration minimizes neuroinflammation and shows anxiolytic, antidepressant and slimming effects in obese rats” has the goal to evaluate the effect of Mucuna pruriens administration on neuroinflammation and behavioural and murinometric parameters in obese rats. The proximate analysis of the chemical composition was performed, and oligosaccharide and phenolic compound profiles of M. pruriens were determined.

Although the interesting theme, some aspects of the introduction, experiments and discussion need considerable improvement.

Answer

We thank the reviewer for the opportunity to improve this manuscript following their comments. We invite the reviewer to see the changes made to the R1 version of the manuscript, especially in the introduction, experiments and discussion sections

Comments

In Introduction section, authors should provide the scientific explanation why the mentioned plant Mucuna pruriens was chosen to establish the targeted activities, as the authors were aware of other activities the mentioned plant was used in Ayurveda. Authors should provide the complete information citing the appropriate literature regarding the traditional use of the investigated plant, not stating only : “…a herbal medicine traditionally used in Ayurveda, but not for slimming purposes...”

Answer

MP was chosen because it has a rich nutritional and phytochemical composition in addition to having hypoglycemic effects (MAJEKODUNMI et al. 2011), antioxidant (TRIPATHI; UPADHYAY, 2002), anti-inflammatory (BALA et al., 2011) and hypocholesterolemic (EZE et al., 2012). However, such effects have been demonstrated in healthy animal models, leaving a gap in scientific knowledge about the effect of MP on an obese model. Thus, both the presence of bioactive compounds and the previously documented effects make MP an adjunctive potential in the treatment of obesity. Based on this hypothesis, we developed this study and reformulated the sentence (please see the sentence page 2- lines 67-76):

“In this context, Mucuna pruriens (L.) DC (Fabaceae) is an herbal medicine traditionally used in Ayurveda which has a rich nutritional and phytochemical composition [11]. Previous studies have demonstrated antioxidant [12], anti-inflammatory [13] and anxiolytic [14] effects in healthy animal models. Such effects associated with the presence of levodopa in Mucuna pruriens (MP) [14,15] can be promising in treating anxiety and depression associated with obesity, since it is precursor of dopamine which act directly in the control of hunger and satiety as well as in promoting well-being [5,16]. However, to our knowledge no study has been developed which has evaluated these effects of MP on obesity with a focus on weight loss and neurobehavioral parameters. In this sense, the present study aimed to evaluate the effect of MP administration on neuroinflammation and on behavioral and murinometric parameters in obese rats.”

Comments

Please, define what part of the plant was used for the tested extracts (legume is not precise terminology, meaning that the author should point if the semen or something else was used).

Answer

We agree with the reviewer. MP seeds were used to prepare the extract, which are the parts of this plant most used in previous studies (MAJEKODUNMI et al., 2012; RAI et al., 2014; RANA; GALANI, 2014). We added this information in subchapter “4.2 Preparation of the MP extract”, please see page 12 – lines 381-392).

References

Majekodunmi S.O.; Oyagbemi, A.A.; Umukoro, S.; Odeku, O.A. Evaluation of the anti-diabetic properties of Mucuna pruriens seed extract. Asian Pac J Trop Med. 2011;4: 632-636.

Rai, S.; Pai, P.G.; Rajeshwari, S.; Ullal, S.D.; Nishith, R.S.; Belagali, Y. Evaluation of Anxiolytic Effect of Chronic Administration of Mucuna Pruriens In Wistar Albino Rats. Am J Pharm Tech Res. 2014;4: 611–619.

Rana, D.G.; Galani, V.J. Dopamine mediated antidepressant effect of Mucuna pruriens seeds in various experimental models of depression. Ayu. 2014;35: 90–97.

Tripathi, Y.B.; Upadhyay, A.K. Effect of the alcohol extract of the seeds of Mucuna pruriens on free radicals and oxidative stress in albino rats. Phytother Res. 2002;16: 534–538.

Bala, V.; Debnath, A.; Shill, A.K.; Bose, U. Anti-Inflammatory, Diuretic and Antibacterial Activities of Aerial Parts of Mucuna pruriens Linn. Int J Pharmacol. 2011;7: 498–503.

Eze, E.D.; Mohammed, A.; Yusuf, K.M.; Tanko, Y.; Sherif, I.A. Changes in lipid profile of rats administered with ethanolic leaf extract of Mucuna pruriens (Fabaceae).  Current Research Journal of Biological Sciences, 2012; 4:130-136.

Comments

Please, give the scientifically explanation for the chosen dose applied in the experiment (750mg/kg)? Why only one dose was used?

Answer

For the selection of the dose administered to the rats, we considered:

  • we conducted a pilot study with doses based on a previous study carried out by Rai et al., (2014) that observed a better anxiolytic effect at the highest doses of MP (500- 750 mg / kg), without adverse effects in healthy rats (without induction of no disease). Considering the 3Rs principle, we select only one dose;
  • we also took into account previous studies that evaluated the toxicity of MP extracts (MANALISHA; CHANDRA, 2012; MUTHU; KRISHNAMOORTHY, 2011) which indicated the use of up to 4000 mg / kg as safe doses of MP;
  • we also consider that MP is normally marketed as an herbal medicine with indication for use at a dose of 800 mg / kg to obtain androgenic effects (AHMAD et al., 2008) and to improve Parkinson's disease (KATZENSCHLAGER et al., 2004).

We insert this information more clearly in the text (please see page 14 - lines 487-489):

References:

Ahmad, M. K.; Mahdi AA; Shukla KK, Islam N, Jaiswar ST, Ahmad S. Effect of Mucuna pruriens on semen profile and biochemical parameters in seminal plasma of infertile men. Fertil Steril. 2008; 90: 627-635.

Katzenschlager, R.; Evans, A.; Manson, A.; Patsalos, P.N.; Ratnaraj, N.; Watt, H.; Timmermann, L.; Van der Giessen, R.; Lees, A.J. Mucuna pruriens in Parkinson’s disease: a double blind clinical and pharmacological study. J Neurol Neurosurg Psychiatry. 2004; 75: 1672-1677

Manalisha, D.; Chandra, K.J. Preliminary phytochemical analysis and acute oral toxicity study of Mucuna pruriens Linn. In albino mice. Int Res J Pharm. 2012; 3: 181-183.

Muthu, K.; Krishnamoorthy. Evaluation of androgenic activity of Mucuna pruriens in male rats. Afr J Biotechnol. 2011;10 (66): 15017-15019

Rai, S.; Pai, P.G.; Rajeshwari, S.; Ullal, S.D.; Nishith, R.S.; Belagali, Y. Evaluation of Anxiolytic Effect of Chronic Administration of Mucuna Pruriens In Wistar Albino Rats. Am J Pharm Tech Res. 2014;4: 611–619.

Comments

The main concerns regarding the presented results refer to the authors’ analysis of the chemical composition of the used extract in the presented and performed experiments. Although the authors based their explanation of the exhibited effects on chemical profile of the extract used in investigation, in Material and Methods they did not provide the procedure of obtaining the extract, the methods used for the qualitative and quantitative chemical analysis of the investigated extracts. It is, at least, insufficient to say that the “lot was the same”, to give the literature from where the methodology was taken over. The subsection 4.1 could not be named….Nutritional characterization of MP, as the data presented in Results showed the presence of non-nutritional secondary metabolites, such as catechin, polyphenolic compounds etc. The whole preparation of the extracts has not been precisely, scientifically and accurately presented. Please, give the HPLC chromatogram, give the brief procedure for extract preparation, and for all methods used to determine the mentioned groups of the compounds present in the extracts. The presented tables in Results section did not give the information whether the presented results refer to plant material or extract (table 1) – please, clarify

Answer

  • We have inserted the subsections entitled “1 Chemicals” and “4.2 Preparation of the MP extract”(please see page11- lines 336-344 and 346-356).
  • The subsection 4.3 has been renamed to “Chemical composition of MP extract” and rewritten with details on the analyses performed (please see pages 12 and 13- lines 394-452).
  • We inform that the chromatograms for the analysis of oligosaccharides (Figure S1), sugars, organic acids (Figure S2) and phenolics (Figure S3) were inserted as complementary materials (please see supplementary materials). Due to the COVID-19 pandemic, it was not possible to access a collaborating institution where the levodopa analysis was performed by chromatography. Thus, we decided to quantify these compounds by NMR in our institution during this week and we sent Figures S4 and S5 (please see supplementary materials) as representative of the results of this analysis.
  • All results presented in Table 1 were obtained from analyses performed on the hydroalcoholic extract of MP powder obtained from the seeds. Please see the new title in Table 1

Comments

Please, explain how the extract was diluted to be administrated in all experiments, taking into account that the extract contained quite different compounds regarding their solubility.

Answer

The MP extract administered to the GSM and GOMP groups was diluted in 1 mL of distilled water, according to Rai et al., (2014), as described on the page 14- lines 484-487. The reviewer's observation is very important and was also our concern. For this reason, before starting the experiment, we observed good solubility of the MP extract (considering the dose 750 mg/kg) in distilled water, despite the presence of quite different compounds regarding their solubility in the MP.

References:

Rai, S.; Pai, P.G.; Rajeshwari, S.; Ullal, S.D.; Nishith, R.S.; Belagali, Y. Evaluation of Anxiolytic Effect of Chronic Administration of Mucuna Pruriens In Wistar Albino Rats. Am J Pharm Tech Res. 2014;4: 611–619.

Comments

Please, give the data whether all animals survived all 8 weeks of the experiment.

Answer

All 32 rats survived until the end of the study, with no adverse events. We added this information in the manuscript (please see page 14 line 493). We inform that we have been working with these gavage protocols and dietary modifications in animal models since 2009 and offer periodic training to the team.

Some minor corrections that should be done:

Comments

English should be corrected: proximal / proximate; sacietogenic / satietogenic: page 2 lines 61, 62; page 3, lines 83,84, page10, line289, should be corrected, and etc…. The Results and Discussion sections should be written in Past Tense. Please, check throughout the text, and prior sending the corrected version, consult the English language native speaker.

Answer

We inform that a native English service have reviewed the entire text, please see below the certificate of review.

Comments

All figures are of poor quality, the quality should be improved.

Answer

We have improved the quality of the figures as per the instructions for authors of Molecules.

Comments

The numbers of subsections should be corrected – there is repeating – 2.3….

Answer

We correct the numbers of subsections. Thank you.

Comments

Page 6, lines 148-156, please, specify the link between the given values and the corresponding investigated groups

Answer

We rewrote the sentence specifying the values in each group (please see page 6 – lines 162-17):

“For the assessment of depressive behavior, the OG group showed less self-cleaning time in the splash test (31.3 s) versus 72.7 in HG, 65.7 in HGMP and 70.07 s in OGMP (p≤ 0.05) (Figure 4). The OG group showed longer immobility (128.1 s) in the forced swimming test versus 57.3 in HG, 69.5 in HGMP and 94.3 s in OGMP (p≤ 0.05) and shorter swimming (116.5 s) versus 198.3 in HG, 148.4 in HGMP and 180.9 s in OGMP (p≤ 0.05) and diving times (1.3 s) versus 21.0 in HG, 25.7 in HGMP and 18.9 s in OGMP (p≤ 0.05) (Figure 5). Animals which exhibit depressive-like behavior in the forced swimming test adopt an immobility posture, while animals which do not exhibit depressive-like behavior remain in motion (swimming or diving). Thus, the longer immobility time in OG indicated depressive behavior in these animals. It is important to highlight that the MP was able to reverse the depressive behavior, since the results were equal to those of the control group.”

Comments

When the plant has been mentioned for the first time, the full Latin name should be provided: Latin name, author abbreviation, and Family, while full name should be in Italic. Afterwards, authors might switch to MP or M. pririens. It refers to Bacterial strains as well (page 8, line 206).

Answer

We correct the words in the text (please see page 2-line 67 and page 9 -line 221).

Comments

The authors should provide the name of the expert who identified the plant material.

Answer

The plant material was identified by botanists Professor Rubens Teixeira de Queiroz (Department of systematics and ecology – Federal University of Paraíba) and Professor José Iranildo Miranda de Melo (Department of Biology, State University of Paraíba). We added this information in the text (please see page 12 -lines 382-383) and in the Acknowledgments section (please page 17 - lines 601-605).

Best regards,

The authors

Reviewer 3 Report

The manuscript:  “Mucuna pruriens administration minimizes neuroinflammation and shows anxiolytic, antidepressant and slimming effects in obese rats” is aimed to evaluate the efect of Mucuna pruriens extracts on neuroinflammation and behaviarol and murinometric parameters in obese rats.

In general the manuscript is well done, presents a documented introduction, well-argued discussions and an appropriate bibliography but, improvements must be made to the results and materials and methods.

The results are well organized but the attached figures are very small and thus their understanding is difficult.

The part of materials and methods needs to be improved: chemicals, chromatographic conditions and extraction procedures, are missing.

Specific comments:

Table 1 - the expression of the quantities was done on the vegetable product?

Line: 24, 63,77- species names must be written in italics

Line 63: please add the family to which the species belongs

Line 335-336: please rephrase the sentence

In subchapter 4.1 - please specify which vegetable product was used

Author Response

João Pessoa/Paraíba - Brazil, november/2020.               

REVIEWER 3

Dear Reviewer,

We are pleased to send you the revised version (R1) of the paper “Mucuna pruriens administration minimizes neuroinflammation and shows anxiolytic, antidepressant and slimming effects in obese rats” after carefully considering the comments of the Reviewers and the Editor. All suggestions and requirements were considered to improve the general scientific quality of the manuscript. All the manuscript was revised in order to fit its format to the standard required by Molecules. The changes in the text are highlighted using the “track changes” function in Microsoft word regarding each Reviewer’s comments. As requested, a clear list of the responses to the Reviewers’ comments (including line numbers that indicate where the changes have been made) is presented.

Comments

The manuscript: “Mucuna pruriens administration minimizes neuroinflammation and shows anxiolytic, antidepressant and slimming effects in obese rats” is aimed to evaluate the effect of Mucuna pruriens extracts on neuroinflammation and behaviarol and murinometric parameters in obese rats.

In general the manuscript is well done, presents a documented introduction, well-argued discussions and an appropriate bibliography but, improvements must be made to the results and materials and methods.

Answer

We appreciate all the reviewer' suggestions and the recognition of the merit of this manuscript. We made changes especially in the Results and Material and Methods sections, as per the reviewer's suggestions.

Comments

The results are well organized but the attached figures are very small and thus their understanding is difficult.

Answer

We have improved the quality of the figures as per the instructions for authors of Molecules.

Comments

The part of materials and methods needs to be improved: chemicals, chromatographic conditions and extraction procedures, are missing.

Answer

We have inserted the subsections entitled “4.1 Chemicals” and “4.2 Preparation of the MP extract” (please see pages 11 and 12- lines 364-379 and 381-392). The subsection 4.3 has been renamed to “Chemical composition of MP extract” and rewritten with details on the analyses performed (please see page 12 and 13- lines 394-452).

Specific comments:

Comments

Table 1 - the expression of the quantities was done on the vegetable product?

Answer

All results presented in Table 1 were obtained from analyses performed on the hydroalcoholic extract of MP seeds (please see page 2- line 77).

Comments

Line: 24, 63,77- species names must be written in italics

Answer

We made these corrections, please see the lines 28, 67 and 91.

Comments

Line 63: please add the family to which the species belongs

Answer

The full Latin name was provided (latin name, author abbreviation, and Family): Mucuna pruriens (L.) DC (Fabaceae) (MP), please see page 2- line 67.

Comments

Line 335-336: please rephrase the sentence

Answer

We rewrote the sentence within subsection “4.2 Preparation of the MP extract” which was added to Material and Methods (please see page 12- lines 381-392).

Comments

In subchapter 4.1 - please specify which vegetable product was used

Answer

MP seeds were used to prepare the extract, which are the parts of this plant most used in previous studies (MAJEKODUNMI et al., 2012; RAI et al., 2014; RANA; GALANI, 2014). We added this information in subchapter “4.2 Preparation of the MP extract”, please see page 12- lines 381-392).

References

Majekodunmi S.O.; Oyagbemi, A.A.; Umukoro, S.; Odeku, O.A. Evaluation of the anti-diabetic properties of Mucuna pruriens seed extract. Asian Pac J Trop Med. 2011;4: 632-636.

Rai, S.; Pai, P.G.; Rajeshwari, S.; Ullal, S.D.; Nishith, R.S.; Belagali, Y. Evaluation of Anxiolytic Effect of Chronic Administration of Mucuna Pruriens In Wistar Albino Rats. Am J Pharm Tech Res. 2014;4: 611–619.

Rana, D.G.; Galani, V.J. Dopamine mediated antidepressant effect of Mucuna pruriens seeds in various experimental models of depression. Ayu. 2014;35: 90–97.

Best regards,

The authors

Round 2

Reviewer 2 Report

Comment on Manuscript: molecules-985934

Recommendation: Accept in present form

The authors improved their manuscript according to the suggested corrections, and thus I recommend the revised version for publication in the Journal.

Reviewer 3 Report

Accept in present form.